# Deregulation of Mitochondrial Calcium Handling Due to Presenilin Loss Disrupts Redox Homeostasis and Promotes Neuronal Dysfunction

**DOI:** 10.3390/antiox11091642

**Published:** 2022-08-24

**Authors:** Kerry C. Ryan, Jocelyn T. Laboy, Kenneth R. Norman

**Affiliations:** Department of Regenerative and Cancer Cell Biology, Albany Medical College, Albany, NY 12208, USA

**Keywords:** Alzheimer’s disease, oxidative stress, presenilin, mitochondria, calcium, neuronal dysfunction, Nrf2

## Abstract

Mitochondrial dysfunction and oxidative stress are major contributors to the pathophysiology of neurodegenerative diseases, including Alzheimer’s disease (AD). However, the mechanisms driving mitochondrial dysfunction and oxidative stress are unclear. Familial AD (fAD) is an early onset form of AD caused primarily by mutations in the presenilin-encoding genes. Previously, using *Caenorhabditis elegans* as a model system to study presenilin function, we found that loss of *C. elegans* presenilin orthologue SEL-12 results in elevated mitochondrial and cytosolic calcium levels. Here, we provide evidence that elevated neuronal mitochondrial generated reactive oxygen species (ROS) and subsequent neurodegeneration in *sel-12* mutants are a consequence of the increase of mitochondrial calcium levels and not cytosolic calcium levels. We also identify mTORC1 signaling as a critical factor in sustaining high ROS in *sel-12* mutants in part through its repression of the ROS scavenging system SKN-1/Nrf. Our study reveals that SEL-12/presenilin loss disrupts neuronal ROS homeostasis by increasing mitochondrial ROS generation and elevating mTORC1 signaling, which exacerbates this imbalance by suppressing SKN-1/Nrf antioxidant activity.

## 1. Introduction

Oxidative stress has emerged as a key driver of many neurodegenerative disorders, including Parkinson’s disease, amyotrophic lateral sclerosis, multiple sclerosis, and Alzheimer’s disease (AD) [1]. The aging nervous system is particularly vulnerable to damage induced by reactive oxygen species (ROS), such as superoxide and hydrogen peroxide [2]. Under normal physiological conditions, the balance between ROS generation and ROS scavenging is tightly regulated. Disturbances to antioxidant scavenging systems or excessive ROS production disrupt many cellular processes and protein homeostasis, contributing to the protein misfolding and aggregation characteristic of neurodegenerative diseases. Mitochondria are major producers of ROS through the byproduct of cellular respiration. Maintaining mitochondrial quality is thus critical for neuronal health and unsurprisingly mitochondrial dysfunction is also implicated in neurodegenerative diseases [3]. Therefore, defining the factors connecting mitochondrial activity and ROS homeostasis is essential to understand the relationship between these systems and neuronal health.

An estimated 55 million people worldwide suffer from dementia, with roughly 70% of these cases resulting from AD (WHO.int). There are currently no effective therapies for AD. Mitochondrial dysfunction and oxidative stress are two key features of AD neurons, but what causes these complications remains unclear [4]. Familial AD (fAD) is an early onset form of the disease resulting from mutations in the presenilin encoding genes, *PSEN1* and *PSEN2*. The presenilins are conserved transmembrane proteins located primarily on the endoplasmic reticulum (ER) and function as the catalytic subunit of the gamma-secretase complex, which cleaves single pass transmembrane proteins [5]. The presenilins have been demonstrated to play an important phylogenetically conserved role in modulating intracellular calcium levels by influencing the rate of ER calcium efflux. Indeed, fAD presenilin mutations or loss of presenilin have been shown to disrupt calcium homeostasis by enhancing ER calcium release [6,7,8,9,10,11,12]. Calcium, as a second messenger, can impact a diverse array of cellular functions and, thus, its role in AD has remained unresolved. Nevertheless, it has been postulated that the resulting elevation in cytosolic calcium and loss of calcium homeostasis due to disrupted presenilin function is a primary contributor to neuronal dysfunction by destabilizing neuronal signaling pathways [13]. In the model system *C. elegans*, it has been demonstrated that loss-of-function mutations in the gene encoding the presenilin ortholog, *sel-12*, result in a rise in mitochondrial as well as cytosolic calcium levels [10,14]. Moreover, the reduction of ER calcium efflux or mitochondrial calcium uptake prevents the premature neurodegeneration phenotypes associated with SEL-12/presenilin loss. Thus, these findings indicate a crucial role of calcium homeostasis in neuronal dysfunction when presenilin function is disrupted. However, the impact that disrupted cytosolic and mitochondrial calcium homeostasis has on ROS production and ROS scavenging is not clear.

Here, to help resolve the relationship between dysregulated neuronal calcium signaling and ROS production and to gain insight into the underlying molecular mechanisms regulating ROS homeostasis, we investigated the contribution of cytosolic and mitochondrial calcium to ROS levels and neurodegeneration. We find evidence that neuronal dysfunction and increased susceptibility to oxidative stress observed in *sel-12* mutants are mediated through the elevation of mitochondrial calcium and not elevated cytosolic calcium levels. Furthermore, we identify an important role of the mTORC1 signaling pathway in exacerbating neurodegeneration and oxidative stress in *sel-12* mutants by sustaining high neuronal ROS levels, which, in part, is likely due to mTORC1 inhibition of SKN-1/Nrf (Nuclear factor erythroid 2-related factor) antioxidant signaling. Overall, our study indicates that SEL-12/presenilin loss disrupts the balance of neuronal ROS by elevating mitochondrial generated ROS and increasing mTORC1 signaling, which in turn disrupts ROS scavenging by inhibiting SKN-1/Nrf.

## 2. Materials and Methods

### 2.1. C. elegans Maintenance and Strains

All *C. elegans* strains were grown at 20 °C on NGM plates seeded with *E. coli* OP50. To age synchronize the animals, gravid worms were bleached, then incubated in M9 for 24–48 h to allow progeny to hatch. These L1 larvae were grown to adulthood on NGM plates. All experiments were performed on day 1 adults.

The following strains were used in the study: N2 was the wild type, *sel-12*(*ar131*) X, *sel-12*(*ty11*) X, *egl-19*(*n2368*) IV, *nprl-3*(*ku540*) IV, *raga-1*(*ok386*) II, *rsks-1*(*ok1255*) III, *sesn-1*(*ok3157*) I, *skn-1*(*lax120*) IV, *skn-1*(*lax188*) IV, *unc-2*(*zf35gf*) X, *bzIs166* [*mec-4p*::mCherry], *dvIs19* [(pAF15)*gst-4p*::GFP::NLS] III, *goeIs22* [*mec-4p*::SL1::GCaMP3.35::SL2::mKate2::*unc-54* 3’UTR + *unc-119*(+)], *jsIs609* [*mec-4p*::MLS::GFP], *uthIs248* [*aak-2p*::*aak-2*(genomic aa1-321)::GFP::*unc-54* 3′UTR + *myo-2p*::tdTOMATO], *zcIs4*[*hsp-4p*::GFP], *zcIs9*[*hsp-60p*::GFP], *zcIs13*[*hsp-6p*::GFP], *zdIs5* [*mec-4p*::GFP + *lin-15*(+)] I, *takEx641*[*mec-7p*::mito-GCaMP6f::SL2::mCherry], and *zhsEx17* [*mec-4p*::MLS::ROGFP]. Genotypes were determined by PCR and DNA sequencing. *sel-12*(*ty11*) mutants were used as the canonical *sel-12* loss of function mutant unless otherwise indicated.

### 2.2. RNAi

The feeding method was used to deliver RNAi [15]. NGM plates were seeded with HT115 bacteria that expressed *skn-1* or *sca-1* double stranded RNAi acquired from the Ahringer library [16], or empty RNAi feeding vector. RNAi bacteria strains were verified by PCR and DNA sequencing. L1 animals were grown on RNAi expressing plates until adulthood.

### 2.3. Analysis of Neuronal Morphology

Animals expressing *mec-7p*::*GFP*(*zdIs5*) were used to examine the structure of the touch receptor neurons (TRNs). Aberrant ALM neurons were scored as either normal or aberrant. Aberrant neurons presented structural defects, such as wave-like bending in the axon, lesions sprouting off the axon, or sprouts stemming from the soma. To image the neurons, worms were immobilized in 100 mM levamisole on 2% agarose pads, and then imaged using the 60x oil objective on a Nikon A1R confocal microscope. The images were compiled using Fiji software. Neuronal morphology was score as healthy by the lack of ectopic sprouts emanating from cell bodies or abnormal projections or gaps in axons. If the observed neuron possessed any abnormal projections, either in the soma or the axon, it was scored as aberrant.

### 2.4. Mitochondrial Organization Analysis

The organization of the mitochondria in the ALM TRN soma was observed in animals expressing *mec-4p*::MLS::GFP(*jsIs609*). Animals were immobilized in 100 mM levamisole on 2% agarose pads, then imaged using 60x oil objective on a Nikon A1R confocal microscope. The images were compiled using Fiji software. Mitochondrial organization was scored as either continuous or discontinuous, where continuous mitochondria had a linear structure without any breaks, while discontinuous mitochondria showed breaks and appeared fragmented. If the observed mitochondria morphology was punctate, it was scored as discontinuous. Alternatively, if the mitochondria morphology was connected, it was scored as continuous.

### 2.5. Paraquat Treatment

Paraquat (Sigma-Aldrich (St. Louis, MO, USA), 856117) was freshly prepared in water and added to NGM agars plates to the indicated final concentration. To induce *hsp-6p*::GFP, and *hsp-60p*::GFP expression, L3 animals were moved to plates containing 2.5 mM paraquat until adulthood and were imaged as day 1 adults. As a control for *gst-4p*::GFP expression, L4 animals were grown on 2 mM paraquat plates overnight and were imaged as day 1 adults. To test *gst-4p*::GFP (*dvIs19*) expression and roGFP1 (*zhsEx17*) sensitivity in wild type and *sel-12* mutants, L3 animals were moved to plates containing the indicated concentration of paraquat (0.01 mM, 0.1 mM, and 1.0 mM) until they reached adulthood (2 days) and were imaged as day 1 adults.

### 2.6. Paraquat Assay

Survival following paraquat exposure was determined by transferring day 1 adults (50 per treatment group) to tubes containing either 0, 50, 100, or 150 mM paraquat for 24 h. Animals were moved to fresh NGM plates and allowed to recover for one hour. Animals were scored as dead if there was no response to repeated prodding with a platinum wire. These experiments were repeated three times.

### 2.7. Mechanosensation Assay

Response to soft touch was performed by using an eyebrow hair attached to a Pasteur pipette, as described [14]. Touches were alternated between the anterior half of the worm (between the pharynx and the vulva) and to the posterior half (between the vulva and tail), for a total of ten touches per worm. A positive response was scored when the animal reversed its forward motion away from the hair and continued moving in the reverse direction. The percentage of positive responses per worm was recorded and then averaged for twenty worms per strain.

### 2.8. Mitochondrial Redox Measurement and GST-4/UPR Reporter Assay

Animals expressing redox sensor roGFP1 targeted to the TRNs (*zhsEx17* [*Pmec-4*::mitoLS::ROGFP]) were immobilized in 100 mM levamisole and mounted on 2% agarose pads on glass slides, then imaged using a 63× oil objective on a Zeiss Axio Observer microscope equipped with an Andor Clara CCD camera. Metamorph software was used to compile images. Samples were sequentially excited with a 405 nm light and 488 nm light with GFP emission detection. Exposure times were kept consistent between samples. The ratio of 405 nm to 488 nm fluorescence intensity was quantified using Fiji software. *gst-4p*::GFP (*dvIs19*) transgenic animals were used as a transcriptional reporter of the SKN-1 target GST-4. *hsp-6p*::GFP (*zcIs13*) and *hsp-60p*::GFP (*zcIs9*) transgenic animals were used as reporters for the UPR^mt^, and *hsp-4p*::GFP *(zcIs4)* animals were used as a reporter for UPR^ER^. For all animals, day 1 adults were imaged using a 10x objective lens on a Zeiss Axio Observer microscope. Exposure times were kept consistent between samples.

### 2.9. Mitochondrial and Cytosolic Calcium Imaging

Mitochondrial calcium was measured in the TRNs in animals expressing *mec-7p*::mito-GCaMP6f::SL2::wrmScarlet. Cytosolic calcium was measured in the TRNs in animals expressing *mec-4p*::SL1::GCaMP3.35::SL2::mKate2. In both cases, animals were immobilized on 100 mM levamisole on 2% agarose pads. Images were taken using a 63X objective lens on a Zeiss Axio Observer microscope equipped with an Andor Clara CCD camera, and images were compiled with Metamorph software (Molecular Devices, version 7.8, San Jose, CA, USA). The fluorescence intensity of mito-GCaMP6 or GCaMP3.35 was normalized to mCherry or mKate2 fluorescent intensity, respectively, as an expression control, and this ratio was quantified using ImageJ (version 1.53s, Bethesda, MD, USA).

### 2.10. Western Analysis

Day 1 adult worms were washed twice in PBS. Half the worm pellet was resuspended in RIPA buffers with protease inhibitors (Roche, Basel, Switzerland), lysed via sonication, and then used to determine protein concentration with a BCA assay (Pierce, Waltham, MA, USA). The other half of the sample was lysed via sonication in 2x Laemmli sample buffer (BioRad, Hercules, CA, USA) containing 5% beta-mercaptoethanol. From this lysate, 20 ug of each sample was loaded and separated with a 10% tris-glycine gel (BioRad). The separated proteins were transferred to a 0.2 μm nitrocellulose membrane (Invitrogen), then incubated in primary antibodies (phospho-Drosophila p70 S6 Kinase (Thr398), 1:500, Cell Signaling #9209, and beta-actin, 1:1000, MP Biomedicals #8691002) in TBS overnight. The membrane was incubated for 1 h in secondary antibodies (IRDye 800CW Goat anti-rabbit (LI-COR), 1:20,000 and IRDye 680RD Goat anti-mouse, 1:20,000 (LI-COR)). LiCor Odyssey CLx infrared imaging system was used to image the blot and the Odyssey Image Studio software (version 5.2, Lincoln, NE, USA) was used to quantify band intensity.

### 2.11. Statisical Analyses

Statistical difference comparing three or more treatment groups was determine using a one-way analysis of variance and a Kruskal–Wallis test used for multiple comparisons of nonparametric data. Non-parametric tests were utilized because all analyses had at least one sample that did not show a normal distribution. Analysis of paraquat data was conducted using a two-way analysis of variance and a Bonferroni correction post hoc analysis. For the mitochondrial and ALM neuronal morphology analyses, a chi-square test was used to determine statistical difference between genotypes. A *p* value of less than 0.05 is considered significant. Graph Pad Prism software (Version 9, CA, USA) was used for all analyses.

## 3. Results

### 3.1. Increase in Mitochondrial Calcium Results in Mitochondrial Redox Imbalance in sel-12 Mutants

Mutations in presenilins have been shown to disrupt calcium signaling in a variety of cell systems. Indeed, evidence in presenilin AD models examining fAD PSEN mutations show enhanced ER calcium release and a rise in cytosolic calcium [7,9,12,17,18]. Consistent with a phylogenetically conserved role of presenilin, mutations in the gene encoding presenilin in *C. elegans*, *sel-12*, also increase cytosolic calcium signaling ([10]; Figure 1A). It has been postulated that the increased release of ER calcium and subsequent rise in neuronal cytosolic calcium is responsible for the profound defects in neuronal function that define fAD [19]. However, the mechanism underlying calcium dysregulation in neurodegeneration is not clear. To investigate the impact cytosolic calcium has on neuronal fitness, we utilized two *C. elegans* mutants that have gain-of-function mutations in the genes encoding the EGL-19 and UNC-2 voltage-gated calcium channels (VGCC). Both the *egl-19*(*n2368gf*) and *unc-2*(*zf35gf*) mutations have been shown to increase the activation state of the EGL-19 VGCC and UNC-2 VGCC, respectively [20,21]. First, to investigate whether these mutations cause elevated cytosolic calcium levels, we employed a cytosolic calcium biosensor, GCaMP3.35::SL2::mKate that is expressed in the touch receptor neurons (TRNs) [22]. The TRNs control the response to soft touch and display distinct age-associated morphological and functional defects, thus providing an attractive system for modeling neurodegeneration [23,24,25]. Consistent with the *egl-19*(*n2368gf*) gain-of-function mutation in *egl-19* increasing the activity of the EGL-19 VGCC, we found a ~4-fold increase in cytosolic calcium in *egl-19*(*n2368gf*) mutants compared to wild type animals (Figure 1A). However, we did not observe an increase in the cytosolic calcium levels in *unc-2*(*zf35gf*) mutants (Figure 1A).

Considering that, in addition to an increase in cytosolic calcium levels, *sel-12* mutants have a significant increase in basal mitochondrial calcium levels that impacts mitochondrial and neuronal function [14,26], we investigated mitochondrial calcium levels in *egl-19*(*n2368gf*) and *unc-2*(*zf35gf*) mutants. Using a mitochondrial calcium biosensor (mito-GCaMP6f::SL2::mCherry) expressed in the TRNs [14,26], we found that unlike the elevated mitochondrial calcium levels observed in *sel-12* mutants, both *egl-19*(*n2368gf*) and *unc-2*(*zf35gf*) mutants were indistinguishable from wild type animals (Figure 1B). These data indicate that a rise in cytosolic calcium levels does not correlate with a rise in mitochondrial calcium levels.

Next, since *sel-12* mutations cause neuronal degeneration as well as mitochondrial morphological abnormalities [10,14], we investigated whether elevated cytosolic calcium results in neuronal or mitochondrial morphological defects. We analyzed the morphology of the ALM TRN soma and axon in transgenic animals expressing soluble GFP within their TRNs. Healthy ALM neurons display round soma and linear axonal processes, whereas aged neurons present neurite sprouts stemming off their soma, and lesions and branching along their axons [14,23,24,25]. Whereas there is a higher frequency of structurally aberrant ALM neurons in *sel-12* mutants, we did not observe structural defects in the age-matched *egl-19*(*n2368gf*) or *unc-2*(*zf35gf*) mutants (Figure 1C,D). To investigate mitochondrial morphology, we examined animals expressing GFP targeted to the mitochondria of TRNs [27]. The mitochondrial network in the soma of wild type animals is continuous and organized in a circular pattern, whereas in *sel-12* mutants it appears discontinuous disorganized and disorganized (Figure 1E,F). Similar to their neuronal morphology, *egl-19*(*n2368gf*) or *unc-2*(*zf35gf*) mutants did not show defects in mitochondrial morphology (Figure 1E,F).

Since mutations in *sel-12* are associated with elevated mitochondrial ROS generation [14], we next examined whether the increased cytosolic calcium observed in *egl-19*(*n2368gf*) mutants disrupts redox homeostasis. To accomplish this, we analyzed animals expressing roGFP1, a redox sensitive GFP, targeted to the mitochondria in TRNs [28]. Oxidation of roGFP1 shifts its peak excitation from 488 to 405 nm [29]. Thus, the 405/488 nm ratio indicates the extent of roGFP1 oxidation. Unlike other ROS biosensors such as HyPer, roGFP1 is independent of pH and thus is a reliable indicator of mitochondrial redox behavior [30,31,32,33] and has been shown to detect oxidative stress in *C. elegans* [34,35]. Consistent with previous observations [14], we found that *sel-12* null mutants had a significant increase in neuronal oxidation, which is rescued by the application of mitoTEMPO, a mitochondrial targeted superoxide scavenger (Figure 1G). In contrast, *egl-19*(*n2368gf*) mutants had neuronal oxidation levels indistinguishable from wild type animals (Figure 1G). Together, these data suggest that the elevated mitochondrial calcium levels observed in *sel-12* mutants and not in *egl-19*(*n2368gf*) mutants is the cause of the elevated oxidative stress and neuronal degeneration observed in *sel-12* mutants. To test this notion, we introduced a null mutation in the mitochondrial calcium uniporter, encoded by *mcu-1*, which we and others have shown reduces mitochondrial calcium uptake in wild type and *sel-12* mutants ([14,36]; Figure 1B). Analysis of roGFP1 fluorescence in *mcu-1*; *sel-12* double mutants demonstrates that reducing mitochondrial calcium uptake prevents the increase in neuronal oxidation observed in *sel-12* mutants (Figure 1G). These data specifically implicate mitochondrial calcium uptake in the increased mitochondrial oxidation levels caused by SEL-12 loss.

Lastly, considering a major function of SEL-12/presenilin is its role as the aspartyl protease subunit of the gamma-secretase complex, we investigated whether *sel-12* mutants that carry a CRISPR/Cas9 induced point mutation, which alters a conserved aspartate residue (D226A) that is required for aspartyl protease activity [37], and have elevated neuronal mitochondrial oxidation. Unlike *sel-12* null mutants, we did not observe an increase in neuronal mitochondrial oxidation in the *sel-12* mutants carrying the D226A mutation (Figure 1G). Thus, the protease activity of SEL-12/presenilin does not impact redox homeostasis as is observed in *sel-12* null mutants. This result is consistent with previous observations that the loss of SEL-12 protease activity does not phenocopy the elevated mitochondrial calcium levels or neurodegeneration observed in *sel-12* null mutants [14,26,37].

### 3.2. Loss of SEL-12/Presenilin Does Not Induce the Mitochondrial Unfolded Protein Response

Previously, it has been demonstrated that an increase in mitochondrial ROS triggers the mitochondrial unfolded protein response (UPR^mt^) [38]. The UPR^mt^ is a phylogenetically conserved adaptive response that functions to maintain mitochondrial proteostasis during mitochondrial dysfunction. Since *sel-12* mutants show disrupted mitochondrial redox homeostasis, we investigated whether *sel-12* mutants have activated UPR^mt^. To accomplish this, we utilized two UPR^mt^ reporters, *hsp-6p*::GFP and *hsp-60p*::GFP [39,40]. As previously shown, using the mitochondrial superoxide inducer, paraquat [41,42], both UPR^mt^ reporters demonstrated robust activity (Figure 2A–C). Surprisingly, despite the increased mitochondrial oxidative status observed in *sel-12* mutants, the *hsp-6p*::GFP and *hsp-60p*::GFP reporter activity in *sel-12* mutants was indistinguishable from wild type animals (Figure 2A-C). These data indicate that the UPR^mt^ is not active in *sel-12* mutants despite the elevated mitochondrial oxidative status.

Since SEL-12, as well as presenilin in mammalian cells, has been shown to be localized to the ER and mediate ER calcium signaling [10,18,43,44,45], we examined the ER unfolded protein response (UPR^ER^) in *sel-12* mutants. Similar to UPR^mt^, UPR^ER^ is a phylogenetically conserved adaptive pathway that maintains ER proteostasis during ER dysfunction. Using the UPR^ER^ reporter, *hsp-4p*::GFP [46], we found no difference in the activity of the UPR^ER^ reporter in *sel-12* mutants compared to wild type animals (Figure 2D). This contrasts with animals treated with *sca-1*(*RNAi*), which knocks down the expression of the sarco-endoplasmic reticulum calcium ATPase and induces a robust UPR^ER^ response (Figure 2D). Together, these results indicate that neither the UPR^mt^ nor the UPR^ER^ are active in *sel-12* mutants.

### 3.3. Inhibition of mTORC1 Improves Mitochondrial Redox Homeostasis and Improves Oxidative Stress Survival in sel-12 Mutants

We previously found that elevated mitochondrial calcium in *sel-12* mutants contributes to neurodegeneration by hyperactivating the mechanistic target of rapamycin 1 (mTORC1) pathway [26]. mTORC1 is a conserved central regulator of cell growth and metabolism. mTORC1 processes a variety of inputs such as growth factors, nutrient signals, and cellular energy status, and in response activates anabolic pathways to promote the production of proteins, lipids, and other biological material [47]. The dysregulation of this pathway has been implicated across multiple diseases, including Alzheimer’s disease, underscoring mTORC1’s importance in regulating cell behavior [48,49]. Inhibiting mTORC1 improves a range of neurodegenerative phenotypes in *sel-12* mutants, including impairments to neuronal and mitochondrial morphology, protein homeostasis, neurodegeneration, and behavior, yet did not reduce mitochondrial calcium levels [26]. Since a rise in mitochondrial ROS production and oxidative stress following mitochondrial calcium uptake mediates the neuronal defects observed in *sel-12* mutants [14,37], we asked if increased mTORC1 activity disrupts mitochondrial redox homeostasis. To answer this, we genetically inhibited mTORC1 in the *sel-12* background by crossing in a null mutation in *raga-1*, which encodes RagA, a GTPase whose activation is required for full mTORC1 activity. We also crossed in a null mutation of *rsks-1*, which encodes ribosomal protein S6 kinase and as an effector of the mTORC1 pathway promotes protein translation. When we compared the oxidation state of roGFP1 in the TRNs between *sel-12* and *raga-1*; *sel-12* mutants, we found that *raga-1*; *sel-12* animals showed a significant reduction in oxidized roGFP1, indicating that mTORC1 inhibition reduces neuronal oxidative stress (Figure 3A). In contrast, *rsks-1*; *sel-12* animals did not show a reduction in oxidized roGFP1 (Figure 3A), suggesting that inhibition of RSKS-1-dependent signaling pathways, such as protein translation through the *rsks-1* deletion, is not sufficient to restore redox homeostasis in *sel-12* mutants, and that mTORC1 signaling affects ROS levels through an alternate mechanism. Additionally, we examined the resistance of these mutants to oxidative stress by exposing the animals to 50, 100, and 150 mM paraquat, which generates superoxide at mitochondria [41,42]. Consistent with the high levels of ROS detected in *sel-12* mutants, we found that *sel-12* mutants have a sharply reduced survival rate compared to wild type animals. In contrast, *raga-1*; *sel-12* animals have an increased survival rate compared to *sel-12* mutants (Figure 3B). However, *rsks-1*; *sel-12* survival was indistinguishable from *sel-12* mutants (Figure 3B). As an alternative strategy to reduce mTORC1 activity, we introduced into the *sel-12* background a constitutively active mutation in the catalytic subunit of 5′ adenosine monophosphate-activated protein kinase (AMPK/AAK-2), a sensor of cellular energy and a major negative regulator of mTORC1 signaling [50]. We also found that the *aak-2*(*ca*); *sel-12* mutants had a significant increase in survival rate compared to *sel-12* mutants when exposed to paraquat (Figure 3B).

We next asked why mTORC1 signaling alters mitochondrial ROS levels in *sel-12* mutants. mTORC1 plays an important role in protein homeostasis by controlling the rate of protein production. Hyperactive mTORC1 may destabilize this homeostasis and increase protein misfolding and aggregation, thereby promoting oxidative stress. However, reducing mTORC1-mediated protein production via the SK6/*rsks-1* mutation was unable to relieve ROS levels in *sel-12* mutants. Evidence in *C. elegans* suggests that mTORC1 signaling additionally influences activation of SKN-1/Nrf in *C. elegans* [51]. SKN-1 is the *C. elegans* orthologue of the Nrf class of transcription factors, which upregulate genes encoding detoxification enzymes to counteract oxidative stress [52]. Importantly, SKN-1 activity is induced by mitochondrial ROS [53,54,55]. Like the Nrf proteins, SKN-1 preserves neuronal health by reducing oxidative stress [56]. The aforementioned study in *C. elegans* by Robida-Stubbs and colleagues showed that mTORC1 inhibition increases transcription of SKN-1 target genes encoding antioxidant proteins [51], implicating mTORC1 as a potential inhibitor of SKN-1/Nrf activity. Thus, we asked whether SKN-1 activity is necessary to improve oxidative stress resistance in *raga-1*; *sel-12* and *aak-2*(*ca*); *sel-12* animals. To this end, we inhibited SKN-1 activity using RNA interference (RNAi) and found that *skn-1* RNAi prevented the increased survival rate in *raga-1*; *sel-12* and *aak-2*(*ca*); *sel-12* animals following paraquat exposure (Figure 3C). We also found that inactivating *skn-1* does not make *sel-12* mutants more susceptible to oxidative stress (Figure 3C), suggesting that *skn-1* is already inhibited in *sel-12* mutants and that this inhibition is relieved by inhibiting mTORC1.

Together, these data indicate that *skn-1* activity is necessary for improvements to oxidative stress resistance following mTORC1 inhibition, in addition to suggesting that *sel-12* mutants have repressed SKN-1 activity. Therefore, we also asked whether the level of SKN-1 activity is altered in *sel-12* mutants. We quantified SKN-1 activity by examining transgenic animals carrying *gst-4p*::GFP, a widely used transcriptional reporter for the SKN-1 transcriptional target and ROS detoxifying enzyme glutathione S-transferase 4 (GST-4) [57]. Surprisingly, despite the higher oxidative stress status in *sel-12* mutants (Figure 1G and Figure 3A–C), *gst-4p*::GFP expression was not induced in these animals (Figure 3D,E), indicating SKN-1 is not being activated by the high mitochondrial ROS levels observed in *sel-12* mutants (Figure 1G and Figure 3A) and implicates the involvement of mTORC1 hyperactivity in SKN-1 repression. Altogether, our results indicate that mTORC1 promotes ROS and oxidative stress sensitivity in *sel-12* mutants, likely, in part, by reducing *skn-1* activity. These data support a model whereby the loss of *sel-12* increases mTORC1 activity, which inhibits *skn-1* function and prevents activation of detoxifying pathways.

### 3.4. sel-12 Mutants Display Reduced SKN-1 Activity in Response to Mitochondrial Oxidative Stress

Since SKN-1 activity is activated by mitochondrial oxidative stress [52] and to further explore SKN-1 activity in *sel-12* animals, we asked if *sel-12* mutants can activate SKN-1 in response to mitochondrial oxidative stress. To induce mitochondrial oxidative stress, we treated *sel-12* mutants with paraquat to induce mitochondrial superoxide production and examined *gst-4p*::GFP expression as a readout for SKN-1 activity. While treatment of wild type animals with 0.01 mM paraquat was able to activate the *gst-4p*::GFP SKN-1 reporter, this concentration only showed baseline *gst-4p*::GFP reporter activity in *sel-12* mutants (Figure 4A). Similarly, treating wild type animals with 0.1 mM paraquat induced a robust increase in *gst-4p*::GFP expression but only a less significant increase of expression was observed in *sel-12* mutants (Figure 4A). In contrast, exposure to 1.0 mM paraquat significantly increased *gst-4p*::GFP activity in *sel-12* mutants. However, this increase was significantly muted compared to wild type animals (Figure 4A). These data suggest that SKN-1 activity is hampered in *sel-12* mutants. Moreover, utilizing the previously described roGFP1 redox biosensor, we found that exposure to 0.1 mM and 1.0 mM paraquat phenocopies the oxidative state observed in *sel-12* mutants (Figure 4B). Taken together, in addition to disrupted mitochondrial redox homeostasis, these data indicate that *sel-12* mutants have a restrained response to oxidative stress, which is consistent with the elevated sensitivity of *sel-12* mutants to oxidative stress (Figure 3B,C).

### 3.5. Activation of SKN-1 Improves Soft Touch Response and Resistance to Oxidative Stress in sel-12 Mutants

We have demonstrated that inhibiting mTORC1 activity is sufficient to reduce the hypersensitivity of *sel-12* mutants to oxidative stress (Figure 3B). Moreover, we showed that this reduction is dependent on SKN-1 function (Figure 3C). Since we have found evidence that SKN-1 activity is restrained in *sel-12* mutants (Figure 4A) and a previous study found that mTORC1 activity inhibits SKN-1 function [51], we next asked whether promoting SKN-1 activation in *sel-12* mutants is sufficient to suppress the neurodegenerative phenotypes or hypersensitivity to oxidative stress observed in *sel-12* mutants. To do this, we crossed in activating mutations in *skn-1* (*lax120* or *lax188*) into the *sel-12* mutant background [58]. We then analyzed defects in the function of the TRNs by measuring the animals’ response to soft touch, a behavior that is controlled by these neurons. Animals with healthy TRNs will reverse their motion and crawl backward when touched on the anterior portion of the body with an eyebrow hair. The reverse will occur when the animal is touched on the posterior half of the body, and the animal will move forward. Aged animals show a reduced frequency in their response rate to soft touch, which correlates with an increased presence of structural abnormalities in the TRNs [24]. Similar to displaying precocious structural ALM neuron abnormalities ([14]; Figure 1C,D), *sel-12* null mutants show defects in soft touch response at day 1 of adulthood ([14]; Figure 5A). In day 1 age-matched adults, we found *skn-1(lax120)*; *sel-12*, or *skn-1*(*lax188*); *sel-12* animals showed a significant increase in soft touch response compared to *sel-12* mutants alone, indicating that increasing *skn-1* activity improves the health of the TRNs in *sel-12* mutants (Figure 5A). Additionally, *skn-1(lax120)*; *sel-12* and *skn-1*(*lax188*); *sel-12* animals showed an increased survival rate following paraquat exposure (Figure 5B) similar to mTORC1 inhibition (Figure 3B). These data indicate that promoting the *skn-1* pathway is sufficient to improve neurodegenerative behavior and enhance oxidative resistance following SEL-12 loss.

### 3.6. Hyperactivation of mTORC1 Is Not Sufficient to Cause Neurodegeneration

We have previously demonstrated that mTORC1 signaling is hyperactivated in *sel-12* mutants and is a driver of neurodegeneration in *sel-12* mutants [26]. Thus, since it was previously shown that increased ROS is a critical contributor to the neurodegeneration observed in *sel-12* animals [14], and we have found that mTORC1 hyperactivity also contributes to the elevated ROS and neurodegeneration in *sel-12* mutants ([26]; Figure 3A), we asked whether the ultimate cause of neurodegeneration in *sel-12* mutants is the hyperactivation of the mTORC1 signaling pathway in *sel-12* mutants. To test this, we examined two mutants that have loss-of-function mutations in two negative regulators of mTORC1 signaling activity. These include the gene encoding the Sestrin ortholog, *sesn-1*, and the gene encoding a protein in the GATOR1 complex, *nprl-3* [59]. While loss of *nprl-3* function has been shown to activate mTORC1 signaling activity in *C. elegans* [60], the impact of *sesn-1* inactivation on mTORC1 activity has not been investigated in *C. elegans*. Thus, to determine whether loss of *sesn-1* results in the activation of mTORC1 signaling activity, we examined the levels of phosphorylated RSKS-1/s6 kinase, a downstream target of mTORC1 signaling, in these mutants [26,61]. Consistent with NPRL-3 and SESN-1 acting as negative regulators of mTORC1 signaling, we found that *nprl-3* and *sesn-1* mutants had significantly elevated phosphorylated RSKS-1 compared to wild type animals, indicating higher mTORC1 activity in these animals compared to wild type animals (Figure 6A,B). However, unlike *sel-12* mutants, when we examined soft touch response in these animals, they were phenotypically wild type, without defects in their response rate (Figure 6C). Additionally, there were no changes to mitochondrial ROS levels in the ALM TRNs of *sesn-1* and *nprl-3* mutants (Figure 6D), nor were there structural defects present within these neurons (Figure 6E). Together, these data indicate that, although mTORC1 plays a crucial role in mediating several defects associated with loss of SEL-12, hyperactive mTORC1 is not on its own sufficient to cause neurodegeneration. Rather, elevated mTORC1 activity in *sel-12* mutants exacerbates the neurodegeneration through its influence on ROS and detoxification pathways via SKN-1.

## 4. Discussion

Oxidative stress increases with age and is a prominent risk factor for many neurodegenerative diseases including AD. Under normal physiological conditions, ROS can act as important signaling molecules, but the aging process and other pathological insults gradually cause a breakdown in ROS scavenging systems and the dysregulation of mitochondrial activity, which further promotes the accumulation of ROS, cellular damage, and impairment of processes governing ROS homeostasis [4]. Calcium signaling plays an important role in ROS generation, and calcium signaling dysregulation is a significant contributor to brain pathology. Specifically, elevated intracellular calcium concentration has been associated with increased oxidative stress in AD models [26,62,63]. Neurons are selectively vulnerable to oxidative stress, and their crosstalk with supporting glial cells is disrupted by redox imbalances [64]. It has been shown that calcium-mediated ROS generation in astrocytes impairs their antioxidant function, which in turn damages neighboring neurons [65]. fAD, which is caused predominantly by mutations in the presenilin encoding genes, is also associated with altered calcium signaling [7,8,11,12]. Presenilin is a transmembrane protein found on most endomembranes and localizes primarily to the ER. Presenilin is known to play an important role in calcium homeostasis. However, the mechanism underlying calcium dysregulation and its relationship to neurodegeneration and oxidative stress is unclear.

Here, our study identified a specific role for mitochondrial calcium in the neurodegeneration induced by presenilin/SEL-12 loss. We found that an elevation of cytosolic calcium, although a consequence of SEL-12 dysfunction [10] as it is with presenilin fAD mutations, is not sufficient to cause neurodegeneration or increase ROS production in *C. elegans* neurons. Rather, an increase in mitochondrial calcium is the critical mediator of ROS generation and neurodegeneration observed in *sel-12* mutants. Increased ROS has severe consequences on brain cell function, damaging their mitochondria and, in more complex organisms, increasing inflammation to further drive ROS production [66]. We also found evidence that the mTORC1 signaling pathway is involved with the increase in ROS levels and oxidative stress hypersensitivity observed in *sel-12* mutants. mTORC1 is a key regulator of cell metabolism, as it senses an array of signals related to nutrients, cellular energy status, and growth signals, and in turn promotes cell growth and biosynthesis pathways. The dysregulation of the mTORC1 pathway has drastic effects on cell behavior and consequently has been shown to be involved in a variety of pathologies [49]. There is evidence that mTORC1 signaling is hyperactivated in AD, including in AD patient brains [67] and in fAD mouse models [68,69]. We have previously found that loss of SEL-12 leads to increased mTORC1 activity associated with mitochondrial calcium signaling and elevated mitochondrial activity [26]. Along with this, we found hyperactive mTORC1 contributes to neuronal defects through its well-described regulation of autophagy. Here we discovered a crucial and less well-studied role for mTORC1 in exacerbating oxidative stress. Although mTORC1’s role in cell metabolism and growth is well known, its relationship to oxidative stress and ROS homeostasis is complicated and far less understood. Studies have shown that hyperactive mTORC1 signaling can promote oxidative stress, leading to cellular damage and disease [70,71,72], including in aging cells [73]. Conversely, rapamycin can confer neuroprotection by reducing oxidative stress [74,75]. We have further implicated a role for mTORC1 in the elevation of mitochondrial ROS levels caused by loss of SEL-12 function.

It is speculated that mTORC1’s promotion of protein production and inhibition of autophagy may promote ROS production by destabilizing protein homeostasis, resulting in increased oxidative stress due to a buildup of protein biomass and misfolded proteins [76]. Interestingly, unlike inhibiting mTORC1 activation (e.g., *raga-1* mutation), we found that inhibition of protein translation via a SK6/*rsks-1* loss-of-function mutation was not sufficient to reduce ROS levels in *sel-12* mutants (Figure 3A). In *C. elegans*, mTORC1 signaling has also been shown to inhibit SKN-1/Nrf activity [51]. SKN-1 is the *C. elegans* orthologue of the Nrf class of proteins, whose best-studied member, Nrf2, is a master regulator of the antioxidant response through its transcriptional upregulation of genes involved in cellular detoxification [52]. Like the Nrf proteins, SKN-1 is important for lifespan extension, and its role in oxidative stress resistance is functionally conserved [77]. SKN-1 activation has been shown to be neuroprotective by reducing oxidative stress [56]. In the aforementioned study linking mTORC1 signaling to reduced SKN-1 activity, it was shown that rapamycin, a specific inhibitor of mTORC1, promotes SKN-1-dependent resistance to oxidative stress [51]. Rapamycin has also been shown to increase Nrf2 activation in cultured human fibroblasts, thereby increasing resistance to oxidative stress and cell lifespan through the delay of replicative senescence [78]. In the present study, we found that the resistance of *sel-12* mutants to oxidative stress through mTORC1 inhibition was a SKN-1 dependent phenomenon. These data suggest that mTOR inhibitors such as rapamycin may aid AD patients by inhibiting neuronal ROS and activating antioxidant systems. The mechanistic connection between mTORC1 signaling and SKN-1/Nrf activation remains an important avenue for further study.

The presenilins’ relationship to fAD has been best studied in their role as the catalytic subunit of the gamma secretase complex, which is involved in the generation of amyloid beta peptides through processing of the amyloid precursor protein (APP). Initial presenilin data provided support for the amyloid cascade hypothesis, which postulates that AD is caused by the pathological build-up of amyloid beta peptides forming plaques in the brain. However, efforts to treat AD by reducing amyloid plaque load have been clinical failures, with immunotherapies targeting amyloid beta showing no effect on disease progression [79,80]. Recent data have indicated AD etiology is more complex and is influenced by additional factors, including dysregulated neuronal calcium signaling [81]. It is interesting to note that presenilin’s function in calcium signaling regulation is gamma-secretase independent [82]. This points to other important neuroprotective roles for presenilin. In this study, we determined that an elevation of mitochondrial calcium specifically is critical for the disruption of redox homeostasis, and previous studies demonstrated that this redox imbalance promotes the loss of proteostasis and neurodegeneration in *sel-12* mutants [14,37]. It is unlikely that amyloid beta peptides are produced in *C. elegans* [83,84], further indicating crucial roles for presenilin independent of amyloid beta peptide generation, especially regarding calcium signaling and ROS homeostasis.

We determined that elevated cytosolic calcium is not sufficient to promote ROS levels and neurodegeneration. We manipulated cytosolic calcium levels through gain-of-function mutations in two voltage-gated calcium channels: *egl-19* and *unc-2*. However, unlike *egl-19*(*gf*) mutants, we did not observe elevated cytosolic calcium in *unc-2(gf)* mutants, despite a previous report showing that this mutation increases calcium currents and causes hyperactive animal behavior [20]. Additionally, we found no structural indications of either neurodegeneration or mitochondrial dysfunction in these mutants (Figure 1C–F). *unc-2* encodes a subunit of the CaV2.1 VGCC, which acts at the presynaptic terminals to promote synaptic transmission [85]. It might be possible that the UNC-2 VGCC does not increase overall cytosolic calcium in the neuron but controls the function of the synapse through localized calcium influx at the presynaptic terminal. Together, these data suggest that subcellularly localized calcium signaling may cause the mitochondrial and neuronal phenotypes observed in *sel-12* mutants. In agreement with this notion, it was previously shown that reducing ER calcium release in *sel-12* mutants was able to restore normal mitochondrial and neuronal function in *sel-12* mutants [10,14].

In congruence, using multiple model systems, the loss of presenilin function has been found to increase ER calcium release into the cytoplasm [7,8,10,11,12]. Moreover, we have found that the increase in mitochondrial calcium observed in *sel-12* mutants is dependent on ER calcium release [14]. However, it is unknown mechanistically how loss of presenilin leads to increased mitochondrial calcium. Interestingly, presenilins are concentrated on the ER that is closely associated with the mitochondria [43]. fAD mutations have also been shown to increase ER-mitochondrial contact sites as well as increase the signaling between the ER and mitochondria and the exchange of metabolites and ions, including calcium [86,87]. However, it is unclear how presenilin alters communication between the ER and the mitochondria, and this is a future direction worth pursuing.

There is an interesting and complex relationship between ROS and longevity. Evidence in *C. elegans* indicates that small increases in superoxide generation may promote longevity by activating the mitochondrial stress response, which upregulates the SKN-1-mediated stress response [88,89,90]. However, *sel-12* mutants have reduced, not increased longevity [10]. Notably, we do not find that the increased ROS we observe in *sel-12* mutants triggers any mitochondrial stress response (Figure 2A–C), nor does it increase SKN-1 activity (Figure 3D,E). This suggests that the pathological nature of SEL-12 dysfunction precludes a protective response to mitochondrial ROS generation.

## 5. Conclusions

Altogether, our study highlights the critical role of mitochondrial calcium in disrupting redox homeostasis and promoting neurodegeneration in animals lacking SEL-12/presenilin function. Additionally, we identified mTORC1 signaling as a mediator of increased mitochondrial ROS, and discovered that its effects on ROS levels are, in part, due to the inhibition of SKN-1/Nrf function. However, the elevation of mTORC1 signaling alone is not sufficient to cause neurodegeneration (Figure 6). It is interesting to note that the inhibition of mTORC1 activity does not alter mitochondrial calcium levels in *sel-12* mutants [26], suggesting both that elevated mTORC1 signaling lies downstream of mitochondrial calcium uptake and that mitochondrial calcium causes additional cellular defects beyond elevating mTORC1 signaling. Overall, our study underscores the specific role of mitochondrial calcium in the neurodegeneration induced by impairment to presenilin/SEL-12 function.

## Figures and Tables

**Figure 1 antioxidants-11-01642-f001:**
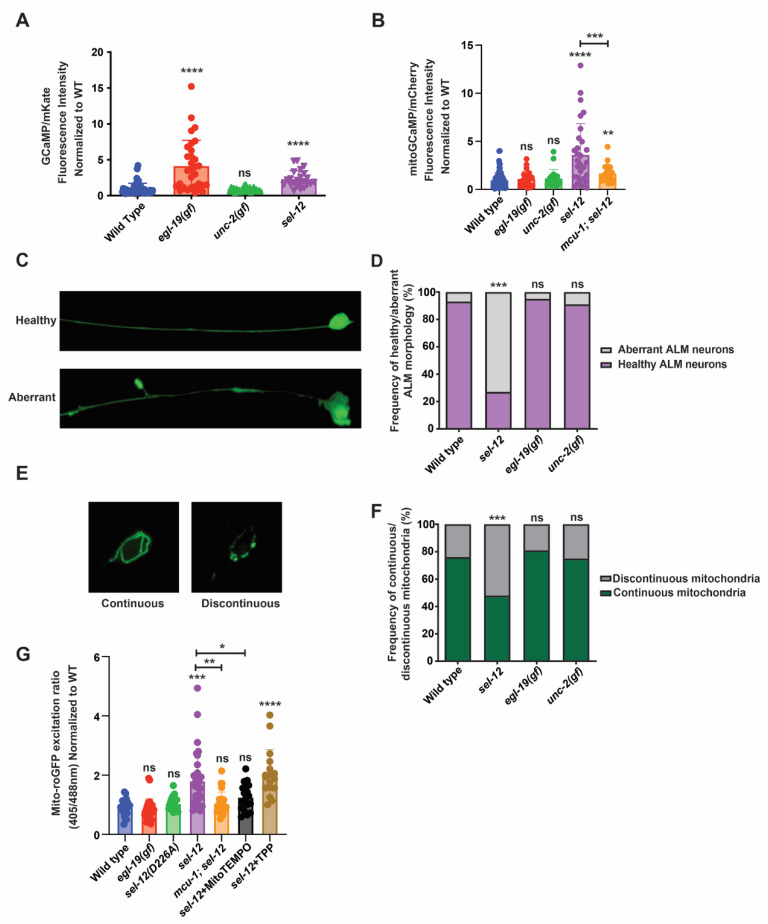
Mitochondrial and neuronal abnormalities in *sel-12* mutants are mediated through mitochondrial calcium. (**A**) Quantification of cytoplasmic calcium levels in animals expressing GCaMP, a genetically encoded calcium biosensor, and mKate, an expression control, in the TRNs. (n ≥ 25). (**B**) Quantification of mitochondria calcium levels in animals expressing mitochondrial-targeted GCaMP6 and mCherry, an expression control, in the TRNs. (n ≥ 21). (**C**) Representative image of healthy and aberrant ALM neurons (scale bar = 10 µm) and (**D**) quantification of the frequency of aberrant or health neurons. (n ≥ 20). (**E**) Representative image of continuous and discontinuous mitochondrial organization in the ALM soma (scale bar = 10 µm) and (**F**) quantification of the frequency of continuous and discontinuous mitochondria. (n ≥ 20). (**G**) Ratio of oxidized to non-oxidized roGFP1 in the mitochondria of ALM neurons as a quantitative measure of oxidation levels. (n ≥ 20). ns *p* > 0.05, * *p* < 0.05, ** *p* < 0.01, *** *p* < 0.001, **** *p* < 0.0001 using one-way ANOVA with Kruskal-Wallis multiple comparison test (**A**,**B**,**G**) or chi-square test (**D**,**F**). Comparisons are made to wild type unless otherwise indicated. Error bars indicate mean +/− SEM.

**Figure 2 antioxidants-11-01642-f002:**
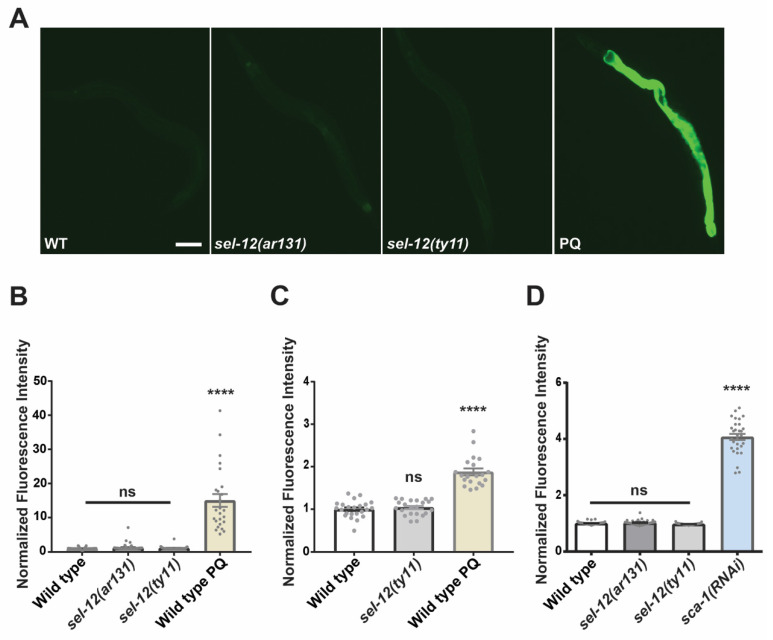
SEL-12/presenilin loss does not induce the mitochondrial unfolded response. (**A**) Representative images (scale bar = 0.1 mm) of GFP fluorescence intensity *hsp-6p*::GFP transgenic animals as a reporter for UPR^mt^. (**B**,**C**) Quantification of GFP fluorescence intensity in *hsp-6p*::GFP (**B**) or in *hsp-60p*::GFP (**C**) transgenic animals as reporters for the UPR^mt^. Paraquat (PQ) was used to induce mitochondrial ROS as a positive control. (**D**) Quantification of GFP fluorescence intensity in *hsp-4p*::GFP transgenic animals as a reporter for UPR^ER^. *sca-1*(*RNAi*) was used as a positive control to knock down expression of the sarco-endoplasmic reticulum calcium ATPase (SERCA) to induce an ER stress response. (n ≥ 20). ns *p* > 0.05, **** *p* < 0.0001 using one-way (**B**,**C**) with Kruskal-Wallis multiple comparison test. Comparisons are made to wild type unless otherwise indicated. Error bars indicate mean +/− SEM.

**Figure 3 antioxidants-11-01642-f003:**
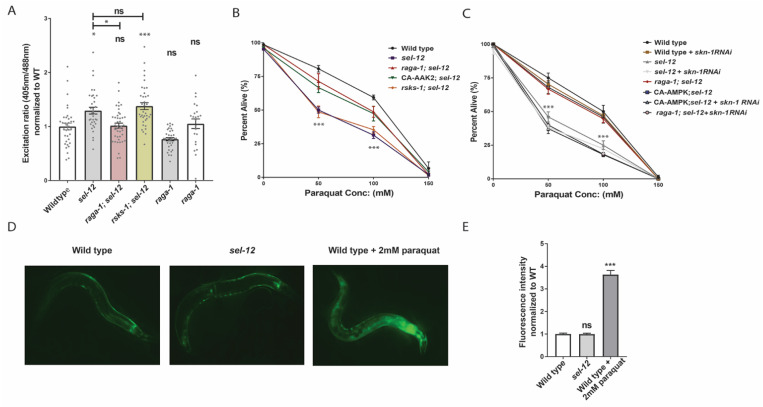
Inhibition of mTORC1 reduces mitochondrial ROS and rescues the hypersensitivity of *sel-12* mutants to oxidative stress. (**A**) Relative mitochondrial oxidation levels in animals expressing mitochondria targeted roGFP1 in the TRNs. (n = 30). (**B**) Survival rate of animals following 24-h exposure to the oxidant paraquat (0, 50, 100, and 150 mM paraquat). (50 animals per strain, performed 3 times) (**C**) Paraquat survival curve in animals exposed to *skn-1* or control RNAi. (**D**,**E**) Representative images (**D**) and quantification (**E**) of GFP fluorescence intensity in *gst-4p*::GFP transgenic animals as a reporter for transcription of SKN-1/Nrf2 target GST-4. ns *p* > 0.05, * *p* < 0.05, *** *p* < 0.001 using one-way ANOVA with Kruskal-Wallis test (**A**,**D**,**E**) or two-way ANOVA with Bonferroni test (**B**,**C**). Comparisons are made to wild type unless otherwise indicated. Error bars indicate mean +/− SEM.

**Figure 4 antioxidants-11-01642-f004:**
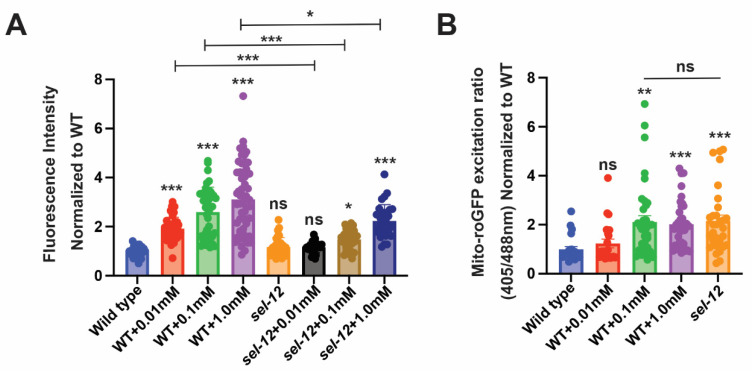
*sel-12* mutants have decreased SKN-1 activity in response to oxidative stress. (**A**) Quantification of *gst-4p*::GFP expression in animals treated with paraquat. (n ≥ 28). (**B**) Ratio of oxidized to non-oxidized roGFP1 in the mitochondria of ALM neurons of animals treated with paraquat. (n ≥ 22). Ns *p* > 0.05, * *p* < 0.05, ***p* < 0.01, *** *p* < 0.001 using one-way ANOVA with Kruskal-Wallis test. Comparisons are made to wild type unless otherwise indicated. Error bars indicate mean +/− SEM.

**Figure 5 antioxidants-11-01642-f005:**
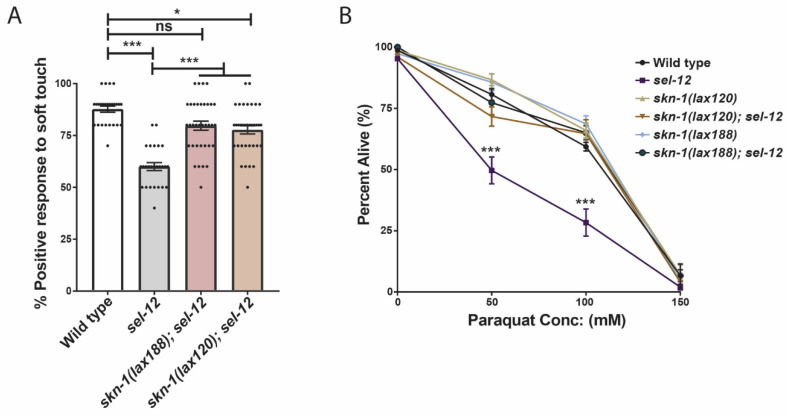
Activation of SKN-1 improves soft touch response and resistance to oxidative stress in *sel-12* mutants. (**A**) Response to anterior and posterior soft touch in *sel-12* mutants and *sel-12* mutants carrying activating mutations in *skn-1*. (n = 20) (**B**) Paraquat survival assay in wild type, *sel-12* mutants, and animals carrying activating mutations in *skn-1* after 24-h exposure to either 50, 100, or 150 mM paraquat (50 animals per strain, performed 3 times). ns *p* > 0.05, * *p* < 0.05, *** *p* < 0.001 using one-way ANOVA with Kruskal-Wallis test (**A**) or two-way ANOVA with Bonferroni test (**B**). Error bars indicate mean +/− SEM.

**Figure 6 antioxidants-11-01642-f006:**
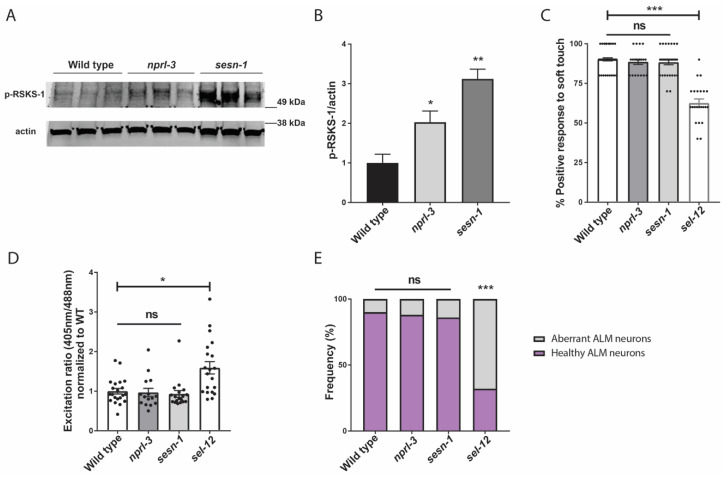
Hyperactivation of mTORC1 is not sufficient to cause neurodegeneration. (**A**) Western blot of p-RSKS-1/S6k in wild type, *nprl-3* and *sesn-1* mutants indicating increased mTORC1 activity and (**B**) quantification of p-RSKS-1/actin in (**A**). (**C**) Quantification of response to anterior and posterior soft touch in wild type, *sesn-1*, *nprl-3*, and *sel-12* mutants. (n = 20). (**D**) Relative mitochondrial oxidation levels in animals expressing mitochondria targeted roGFP1 in the TRNs. (n ≥ 15). (**E**) Quantification of frequency of healthy and aberrant ALM neurons present in wild type, *sesn-1*, *nprl-3*, and *sel-12* mutants. (n ≥ 20). ns *p* > 0.05, * *p* < 0.05, ** *p* < 0.01, *** *p* < 0.001 using one-way ANOVA with Kruskal-Wallis multiple comparison test. Comparisons are made to wild type unless otherwise indicated. Error bars indicate mean +/− SEM.

## Data Availability

The data collected and used for this study are presented in the manuscript and reagents are available from the corresponding author upon request.

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
