# Peer review of "Deregulation of Mitochondrial Calcium Handling Due to Presenilin Loss Disrupts Redox Homeostasis and Promotes Neuronal Dysfunction"

_antioxidants, 2022, doi:10.3390/antiox11091642_

Round 1
Reviewer 1 Report
The manuscript is well writed and its distribution is ok. The information is interesting but major revision will be necesary.
1) Calcium signaling plays an important role in ROS generation, as author indicated, but they are speaking all time about neurons. But, what happen with other cells, such as astrocytes. These kind of neural cells use calcium to communicate with another astrocytes, with neurons and other cells inside central nervous system. Authors need to discuss it.
2) Authors worked in presenilin. Amyloid beta is eliminated thanks to presenilin but, actually the hipotesis of amylid is not approved for many scientifics. In discussion paragraps must be appear this point of view.
3) Authors did not indicate nothing about inflammation, because this an important point in mitochondria affectation.
4) Respect the western-blot, authors need to present all the gel with indication of the kDa of each band. This can be added in other extre-document.
5) Some of the conclusions are to much strong for the results indicated. Please you must be more concret
6) Some words are bad write
7) With all the above, I consider that the article is very interesting, although many of the results have not been positive
Author Response
We would like to thank the reviewer for their constructive criticisms. We have incorporated the changes suggested by the reviewer, which we feel has improved the manuscript. Below we address the reviewer’s comments:
“1) Calcium signaling plays an important role in ROS generation, as author indicated, but they are speaking all time about neurons. But, what happen with other cells, such as astrocytes. These kind of neural cells use calcium to communicate with another astrocytes, with neurons and other cells inside central nervous system. Authors need to discuss it.”
We have added the importance that calcium and ROS have on astrocytes function and the impact this has neuronal function to the discussion (lines 514-516).
“2) Authors worked in presenilin. Amyloid beta is eliminated thanks to presenilin but, actually the hipotesis of amylid is not approved for many scientifics. In discussion paragraps must be appear this point of view.”
We have added discussion to address this important point (lines 573-579).
“3) Authors did not indicate nothing about inflammation, because this an important point in mitochondria affectation.”
On lines 527-529, we now include discussion of the impact mitochondrial ROS can have on inflammation.
“4) Respect the western-blot, authors need to present all the gel with indication of the kDa of each band. This can be added in other extre-document.”
We have added the molecular markers to the western blot image.
“5) Some of the conclusions are to much strong for the results indicated. Please you must be more concret”
Since we were not confident what the reviewer was referring to, we did not address this point.
“6) Some words are bad write”
We have carefully proofread the manuscript and fixed typos.
Reviewer 2 Report
Norman group report a nice contribution for the critical role of mitochondrial calcium in disrupt- 622 ing redox homeostasis and promoting neurodegeneration in animals lacking SEL-12/pre- 623 senilin function. This manuscript is concise, well written and presents an interesting piece in this field. I have no claim.
Author Response
We appreciate the response of the reviewer and thank them for carefully reviewing our manuscript.
“Norman group report a nice contribution for the critical role of mitochondrial calcium in disrupt- 622 ing redox homeostasis and promoting neurodegeneration in animals lacking SEL-12/pre- 623 senilin function. This manuscript is concise, well written and presents an interesting piece in this field. I have no claim.”
Reviewer 3 Report
The purpose of this manuscript is to study the mechanism that regulates mitochondrial dysfunction and oxidative stress linked to neurodegenerative diseases and in particular in Alzehimer's. One of the gene involved in the family related form of Alzehimer's is the presenilin gene whose mutations are responsible for an altered homeostasis of calcium in the endoplasmic reticulum. Therefore, the authors want to study the relationships between the alteration of calcium with consequent neuronal dysfunction and the production of ROS responsible for neurodegeneration. To do this the authors use C. elegans as an animal model , widely used for studies in SNC and investigate a series of pathways and cell signals correlated with the function of presenilin gene.
The manuscript is well structured, the figure are of satisfactory resolution and the results confirm the role of presenilin loss in the increase of ROS production correlated to mitochondrial calcium levels. On the other hand, the authors demostrate the role of mTORC as a critical signal factor in supporting the ROS levels through repression of the SKN1/Nrf scavenger.
However, some aspects of the manuscript need to be addressed.
Minor revisions: please standardize the characters of cellular strains; adapt the references indicating the authors of the articles cited;
Do the authors checked the levels of cytosolic and mitochondrial expression of the calcium biosensors in neurons?; how do they say that in all the animals tested the levels of expression are comparable? have the authors tried to make a determination of cytosolic and mitochondrial calcium levels by isolating cell organelles? Reduce par 3.1 and 3.3 because they are too long and dispersive.
Author Response
We would like to thank the review for carefully reviewing and appreciate their comments to improve our manuscript. Below we address the reviewer’s comments:
“Minor revisions: please standardize the characters of cellular strains; adapt the references indicating the authors of the articles cited”
We thank the review for identifying this inconsistency. This has been fixed.
“Do the authors checked the levels of cytosolic and mitochondrial expression of the calcium biosensors in neurons?;”
Our calcium biosensors are tandemly expressed with RFP. For example, we use a TRN promotor (5’ regulatory region) to tandemly express GCaMP and RFP (e.g., mec-4p::GCaMP3.35::SL2::mKate). The cytosolic reporter expresses both GCaMP3.35 and mKate. mKate is used for normalizing expression. The mitochondrial reporter expresses mito-GCaMP6f::SL2::mCherry. mCherry is used for normalizing expression. The calcium reporter data is presented as a ratio of GCaMP/RFP expression. These methods are described on lines 157-166 and details are added to Figure 1’s legend for further clarity.
“have the authors tried to make a determination of cytosolic and mitochondrial calcium levels by isolating cell organelles?”
We have not analyzed isolated organelles. Our data indicates that the increase in mitochondrial calcium likely occurs by ER to mitochondrial calcium signaling so isolating mitochondria would disrupt the signaling between these two organelles.
“Reduce par 3.1 and 3.3 because they are too long and dispersive.”
To be more concise, we have removed some or shortened sentences in the indicated the sections.
Reviewer 4 Report
The article could be published as it is, but first, the authors should make small changes to improve the manuscript.
1) Introduction: In the introduction, it should be listed or explain the neurodegenerative disorders in which oxidative stress is involved. It would be interesting to provide some data about the global burden of AD or the prevalence in developed countries.
2 2). The authors have compared the student's t-test and performed one-way ANOVA with a posteriori or post hoc contrasts. These tests are parametric. The authors should evaluate whether the data have a normal distribution. If they do not have it, they should use non-parametric tests, such as Mann-Witney U and Kruskal Wallis test. In any case, the following sentence should be included. The normality of the data was evaluated, and the variables were analyzed using: ANOVA and t-tests, Kruskal–Wallis or their non-parametric equivalents Mann–Whitney U-tests. Both normality and non-parametric tests can be quickly done with the GraphPad software used by the authors.
3) In the discussion should, it should be indicated that future research should be conducted after the results of your study.
4 4) Finally Also, comment in the discussion on the possible preventive or therapeutic implications of the results of this study
Author Response
We would like to thank the review for carefully reviewing and making the suggestion to re-evaluate using a non-parametric vs parametric statistical analyses, which has improved the rigor of our study.
“1) Introduction: In the introduction, it should be listed or explain the neurodegenerative disorders in which oxidative stress is involved. It would be interesting to provide some data about the global burden of AD or the prevalence in developed countries.”
Agreed, oxidative stress is involved in many neurodegenerative disorders. We have added this change to lines 26-28. Also, the AD prevalence has been added to lines 39-41.
“2). The authors have compared the student's t-test and performed one-way ANOVA with a posteriori or post hoc contrasts. These tests are parametric. The authors should evaluate whether the data have a normal distribution. If they do not have it, they should use non-parametric tests, such as Mann-Witney U and Kruskal Wallis test. In any case, the following sentence should be included. The normality of the data was evaluated, and the variables were analyzed using: ANOVA and t-tests, Kruskal–Wallis or their non-parametric equivalents Mann–Whitney U-tests. Both normality and non-parametric tests can be quickly done with the GraphPad software used by the authors.”
We have checked our datasets and have found that at least one sample in each analysis shows a non-normal distribution so we used non-parametric test for these datasets. This is now indicated in our figures and methods (line 181-184).
“3) In the discussion should, it should be indicated that future research should be conducted after the results of your study.”
On lines 569-570, we state that the mechanistic connection between mTORC1 signaling and SKN-1/Nrf needs further investigation. Also, on lines 611-612, we state that the role presenilin has in affecting ER-mitochondrial calcium signaling is not understood and needs for investigation.
“4) Finally Also, comment in the discussion on the possible preventive or therapeutic implications of the results of this study”
Lines 567-569 discuss the possibility of rapamycin as a therapeutic intervention.
Round 2
Reviewer 1 Report
the paper need corrections in some words.
Author Response
We would like to thank the reviewer for identifying typos that are in need of correction. We have thoroughly proofread and spell checked our manuscript and believe we have found all the typos.